# Potential implementation strategies, acceptability, and feasibility of new and repurposed TB vaccines

**Puck T. Pelzer**[1,2]*, **Janet Seeley**[2], **Fiona Yueqian Sun**[2], **Michele Tameris**[3], **Li Tao**[4], **Zhao Yanlin**[4], **Hisham Moosan**[5], **Chathika Weerasuriya**[2], **Miqdad Asaria**[6], **Sahan Jayawardana**[6], **Richard G. White**[2‡], **Rebecca C. Harris**[2¤‡]

**1** KNCV Tuberculosis Foundation, Amsterdam, Netherlands, **2** London School of Hygiene & Tropical Medicine (LSHTM), London, United Kingdom, **3** South African Tuberculosis Vaccine Initiative (SATVI), Department of Pathology, Institute of Infectious Disease and Molecular Medicine, University of Cape Town (UCT), Cape Town, South Africa, **4** Chinese Centre for Disease Control and Prevention, Beijing, China, **5** Health Action by People, Thriuvananthapuram, Kerala, India, **6** London School of Economics (LSE), London, United Kingdom

¤ Current addresss: Sanofi Pasteur, Singapore, Singapore
‡ These authors are joint senior authors on this work.
* puck.pelzer@knctbc.org

**Data Availability Statement:** We cannot share the data as we ensured confidentiality to our interviewees, and the transcripts could traced back to them. As such, we are unable to provide an

## Abstract

Recently, two Phase 2B tuberculosis vaccine trials reported positive efficacy results in adolescents and adults. However, experience in vaccinating these age groups is limited. We identified potential implementation strategies for the M72/AS01$_E$ vaccination and BCG-revaccination-like candidates and explored their acceptability and feasibility. We conducted in-depth semi-structured interviews among key decision makers to identify implementation strategies and target groups in South Africa, India, and China. Thematic and deductive analysis using a coding framework were used to identify themes across and within settings. In all three countries there was interest in novel TB vaccines, with school-attending adolescents named as a likely target group. In China and India, older people were also identified as a target group. Routine vaccination was preferred in all countries due to stigma and logistical issues with targeted mass campaigns. Perceived benefits for implementation of M72/AS01$_E$ were the likely efficacy in individuals with *Mycobacterium tuberculosis* (Mtb) infection and efficacy for people living with HIV. Perceived challenges for M72/AS01$_E$ included the infrastructure and the two-dose regimen required. Stakeholders valued the familiarity of BCG but were concerned about the adverse effects in people living with HIV, a particular concern in South Africa. Implementation challenges and opportunities were identified in all three countries. Our study provides crucial information for implementing novel TB vaccines in specific target groups and on country specific acceptability and feasibility. Key groups for vaccine implementation in these settings were identified, and should be included in clinical trials and implementation planning.

anonymised dataset. However, we are happy to share the data upon request to interested, qualified researchers. As we do not have administrative support to be able to provide a non-author point of contact, the authors will be the points of contact for fielding data access requests (Please contact Richard White (PI and co-last author): richard. white@lshtm.ac.uk, or Puck Pelzer (correspondence author): puck.pelzer@kncvtbc. org). Audio data and transcripts will be stored on a secure server for certain years, and researcher notes will be stored in a locked cabinet. Data will be managed by the project coordinator, under the guidance of the PI. Access to the original data will be limited to members of the study team. Interested researchers will be invited to complete a data sharing agreement, on request to the corresponding author, for access to anonymised transcript excerpts. Although the authors cannot make their study's data publicly available at the time of publication, all authors commit to make the data underlying the findings described in this study fully available without restriction to those who request the data, in compliance with the PLOS Data Availability policy. For data sets involving personally identifiable information or other sensitive data, data sharing is contingent on the data being handled appropriately by the data requester and in accordance with all applicable local requirements.

**Funding:** Work was funded by UKRI funding via SET Bloomsbury CCF17-7779. RGW is funded by the Wellcome Trust (218261/Z/19/Z), NIH (1R01AI147321-01), EDTCP (RIA208D-2505B), UK MRC (CCF17-7779 via SET Bloomsbury), ESRC (ES/P008011/1), BMGF (OPP1084276, OPP1135288 & INV-001754), and the WHO (2020/ 985800-0). The funders had no role in study design, data collection and analysis, decision to publish, or preparation of the manuscript.

**Competing interests:** I have read the journal's policy, and the authors of this manuscript have the following competing interests: Rebecca Harris reports current employment by Sanofi Pasteur; her Sanofi employment includes work on COVID-19, but is unrelated to TB.

## Introduction

Tuberculosis (TB) is a major cause of global adult mortality [1]. There were 10 million incident cases of TB globally in 2019, of which 465,000 were multi-drug resistant, with a current average annual incidence rate decline of only 2%. Adolescents and young adults account for most global tuberculosis patients [1]. These groups are, therefore, target populations for intervention. Currently, there is only one registered vaccine available for TB, Bacillus Calmette Guérin vaccine (BCG) [2]. BCG has been shown to prevent TB (particularly TB meningitis and disseminated TB) in young children, but provides variable protection against pulmonary TB [3]. New tools such as vaccines are urgently needed to accelerate progress towards the World Health Organization 'End TB' and elimination goals, and to prevent further development of multi-drug resistance [4].

Modelling of new TB vaccines suggests that greater and faster impact would be achieved by vaccinating adolescents and adults, instead of children, with new TB vaccines [5, 6], and recently attention has switched from funding trials in infants, to trials in adolescents and adults. In 2018, two phase IIB trials (proof of concept trial involving a larger group of people) reported significant positive efficacy results [7, 8]. A study in HIV-negative adolescents with a negative Interferon-γ release assay (IGRA) at baseline (interpreted as not *Mycobacterium tuberculosis* [Mtb] infected), reported 45% (95% confidence interval [CI]: 6–68%) efficacy for prevention of sustained IGRA conversion (and presumably Mtb infection) among BCG revaccinated adolescents in South Africa [7]. Currently BCG is only recommended for new-borns. BCG is known to be safe when given to IGRA-positive (i.e. infected) populations, but its efficacy in these populations is unknown [9]. BCG is currently contraindicated in people living with HIV (PLHIV). Duration of protection of adolescent vaccination has not been studied, but infant vaccination has demonstrated duration of efficacy of up to 15 years, with some studies suggesting even longer durations are possible [10]. Given that BCG is already indicated for infants, with sufficient supportive evidence for adolescent revaccination, countries could potentially adjust their BCG vaccination policy to include a second dose in adolescence. Consequently, policy makers are discussing the reintroduction of adolescent BCG revaccination.

The second study reported 49.7% (90% CI: 12.1–71.2%) efficacy against TB disease of the M72/AS01$_E$ vaccine candidate among Mtb infected (IGRA positive), HIV-negative, 18–50 year old adults in South Africa, Kenya and Zambia [8]. The vaccine has been shown to be safe and induce an immune response in IGRA-negative populations and HIV-positive populations, but its efficacy in these populations is currently unknown as it has not been explored in clinical trials. This was the first study to present the efficacy of a robust new vaccine candidate since BCG.

These trial results suggest a change to the current vaccination strategy for TB, from infants to adolescents and adults, may be required. This presents challenges, however, as the health system structure for TB vaccine programs is targeted primarily at children and is separate from TB care programs, [11–13] therefore country TB programs have very little experience vaccinating adolescents and adults. There is currently a lack of evidence on potential strategies for adolescent and adult TB vaccination implementation, what could benefit or be a challenge to implementation, and what practical requirements would be.

In our study we aimed to understand potential implementation strategies for the two vaccines, M72/AS01$_E$ and BCG revaccination, and the possible benefits of and challenges to implementation. We conducted interviews with policymakers and key public health or immunization stakeholders in three high TB prevalence countries: India, South Africa, and China. We explored issues around the health care system in each country and identified various country-specific criteria and challenges that would need to be addressed to design a successful

adolescent and adult TB vaccination strategy. Through this process we identified and evaluated a range of possible implementation strategies. Our findings can inform other disease-specific adolescent and adult vaccination programs such as SARS-CoV-2 and HIV.

## Methods

### Study design

As part of a multi-disciplinary research project for the strategic and translational development of late-stage tuberculosis vaccine candidates, we conducted a study among key experts from South Africa, India, and China. Between May and December 2020, we conducted semi-structured interviews to assess context-specific acceptability and feasibility of adolescent/adult TB vaccine implementation. Additionally, we identified country-specific scenarios with high potential for successful TB vaccine implementation. The interviews consisted of two sections: 1) a semi-structured section exploring the context-specific acceptability and feasibility of TB vaccines for adolescents and adults, and 2) a series of questions with regards to possible implementation scenarios.

### Study population and setting

We selected India and China because these countries contribute the first and second largest number of cases to the global annual incidence of TB respectively, and South Africa has one of the highest TB incidence rates [14]. Furthermore, South Africa is a strategically important setting for this vaccine, and a likely early adopter of any new TB vaccine developed, due to strong involvement in the respective phase IIB studies.

Using purposive sampling, we recruited 22 participants, aiming for eight from each country. In India the sample was limited to six due to the unavailability of interviewees during the COVID-19 pandemic. Participants were selected based on their expertise (TB, vaccines, vaccine policy, vaccine supply, vaccine delivery), sector (civil society, academic, Ministry of Health, Ministry of Finance, or non-governmental organization membership) and scope (national, regional, and local level). We sought to cover a wide range of expertise, aiming for two interviewees in each area of expertise, sector, and scope (S1 Table). Prior to the interview, a short summary of the results of the two vaccine trials and other relevant information regarding the format of the interview were sent via email (S1 File).

### Data collection

After piloting the tool, interviews were conducted by a researcher (first author) experienced in conducting semi-structured interviews, and with a good knowledge of TB vaccines and TB. All study participants provided digital informed consent for participation in the study and for audio recording. All data were anonymized and kept securely. Interviews were conducted in English and took approximately one and a half hours and were conducted and audio-recorded through Zoom (Video Communications Inc., 2016). Five of the interviews with Chinese participants were conducted through a Mandarin translator experienced in qualitative research (third author). Audio recordings of the interviews were transcribed, and transcripts were quality-checked by the interviewer and the translated interviews were validated by the translator.

The first section of the interview consisted of semi-structured questions which focused on acceptability, challenges, and solutions (S2 File). In the second section on implementation strategies the interviewee was asked specific questions regarding implementation strategies. When an age group, such as adolescents and adults, was mentioned, the interviewer asked for a suggested age range. The interviewer prompted for answers on the population, age-related

groups, HIV status and socio-economic status (SES), and high priority risk groups first (e.g., drug users, people living in high density areas/slums, miners, diabetics, immunocompromised people, people in chronic care, and TB patients). Where an interviewee had time restrictions, if many target groups were named, the interviewer focused on the groups to which the interviewee assigned highest priority.

This study was approved by the London School of Hygiene and Tropical Medicine Ethics Committee, the University of Cape Town Health Sciences Faculty Human Research Ethics Committee, Indian Council of Medical Research, and the Ethical Review Committee of Chinese Center for Disease Control and Prevention.

## Analysis

For data from the first section of the semi-structured interviews, we used thematic and deductive analysis to tease out themes from the data as well as using a predefined coding scheme based on the interview guide. Analyses were conducted to identify unifying themes across settings and by country to identify setting-specific themes. We used NVivo (QSR International Pty Ltd. Version 12, 2018) to manage the analysis. The themes identified were used to develop a coding framework. Two researchers discussed coding, (sub)themes and interpretation of the data. We presented the study findings from the semi-structured interviews using a narrative descriptive approach. Quotations from the interviews were used to highlight participant views. We labelled the quotations with country and participant's main areas of expertise–for example 'South Africa -vaccines, delivery, finance'. To maintain confidentiality, names and designations were not used. We presented these findings according to benefits, challenges, and requirements per country.

We summarized the implementation scenarios from the second section of the interview in tabular format using Microsoft Excel (Microsoft Corp, Seattle 206 WA, USA). We provided a weighted average for number of times scenarios/strategies that were mentioned by interviewees. The implementation strategies were summarized and described per country, vaccine, and target group, and accompanied by narrative where needed. Where the interviewee did not offer information on strategies or scenarios, this section was left blank.

## Results

A total of 22 interviewees participated in this study: eight each from South Africa and China and six from India. An overview of the expertise for each interviewee is provided in S1 Table. In summary their expertise consisted of: TB, vaccines, vaccine policy, vaccine supply, vaccine delivery); the sector: civil society, academic, Ministry of Health, Ministry of Finance, or non-governmental organization membership; and scope: national, regional, and local level. The majority of the expertise was in TB in general and at the national level.

### Section 1: A semi-structured section exploring the context-specific acceptability and feasibility of TB vaccines for adolescents and adults

Overall themes identified were benefits, requirements, and challenges, which were categorized into acceptability and feasibility. These were further divided into different codes: political, financial, logistical, willingness and vaccine characteristics. Results are presented by country.

**South Africa.** Interviewees anticipated that a new TB vaccine would be readily accepted. They pointed out that because of the high TB and HIV burden in South Africa, there is a strong political will to prioritize TB vaccination implementation within the healthcare system. One respondent commented that familiarity with the BCG vaccine, particularly for health care

workers, made it "*more logical for the government to go with BCG*". (South Africa-TB, vaccines, delivery)

Interviewees reported no strong anti-vaccination movement in South Africa and indicated that there are civil society groups in place which could increase vaccine uptake. In terms of feasibility, interviewees appreciated that the TB vaccine could piggyback on an existing program and could learn from HPV and COVID-19 vaccination strategies:

"*.. so one of the big challenges during COVID-19 has been that the system almost entirely relies on individuals coming into the public health system for vaccinations. So, 80% of the population have to come into the facilities, and what that taught us during COVID-19 is that our assumption was fatally flawed because people don't have transportation, people don't have access [. . .] And if that happens, your vaccination rates drop dramatically.*" (South Africa-vaccines, delivery, finance)

Interviewees drew other parallels with COVID-19 vaccination and reflected on the impact of that vaccination campaign for TB:

"*So, if the COVID-19 vaccine campaign is a failure, it is going to be a big hurdle for the TB vaccine coming after it. If it (the COVID-19 vaccine campaign) is a success, then obviously there's a lot of learnings we can have from that, but I think there are significant community and individual um barriers which need to be overcome.*" (South Africa-TB, TB vaccines, academic)

While there was no strong anti-vaccination movement in South Africa, some interviewees emphasized that coverage and implementation of vaccines will depend on the attitude of the general population. Challenges highlighted in terms of acceptability included the potential reluctance to take up vaccines due to the lack of knowledge and education on TB.

"*There could be a challenge around uptake, people may feel scared or worried to actually go for the vaccination. So, I think stigma could be a concern. [. . .] Historically it hasn't been a major issue in South Africa, but over the last six months, we've certainly seen vaccine hesitancy take a foot hold in South Africa.*" (South Africa-vaccines, delivery, finance)

In addition to potential barriers from the public, interviewees also expressed concerns about the lack of political urgency and willingness. The costs of delivery and procuring vaccines could be a financial constraint and many interviewees worried that there would be no interest in producing TB vaccines, as they are not considered to be profitable:

"*The other issue is (vaccines') manufacturing and availability. If the vaccine does work, is it going to be available to us*?" (South Africa-TB, TB vaccines, academic)

Another challenge for M72/AS01$_E$-like vaccines was the need for a second dose, which was thought could affect the proportion of individuals becoming fully vaccinated as people would not come back for the second dose. Furthermore, interviewees worried that M72/AS01$_E$'s adjuvant could be expensive and scarce. At the same time, it was also suggested that BCG could be scarce.

"*That's the adjuvant and that is in scarce supply. So, it's not just the M72/AS01$_E$ that's important, it's the adjuvant (AS01 $_E$) is in scarce supply at a global level and will probably be reoriented to other vaccines*" (South Africa-vaccines, TB, policy)

The biggest concern raised for both vaccines was required testing for HIV and/or Mtb. This was considered a potential deal breaker because of the logistical and cost barriers it would present.

Most interviewees suggested that addressing hesitancy and engaging civil society could increase coverage, especially in minority groups. Interviewees thought local manufacturing would increase ownership and therefore uptake. A need identified was to set up a proper data system for adverse events following immunization monitoring. Other aspects that would facilitate uptake would be a single dose vaccine, with easy delivery at lower health care level:

"*Simplicity of implementation, so if a single dose could be administered by the lowest level healthcare worker, and isn't reliant on a supply chain, doesn't need multiple doses.*" (South Africa-vaccines, delivery, finance)

In terms of evidence, most interviewees required the vaccine to have safety and efficacy in risk groups such as PLHIV.

"*You want a vaccine that is both broad spectrum and can be applied to all people, right? Because one of the issues in countries like South Africa is equity. So, if you're protecting some people and not others, especially if we decide that, if it's not registered for use in HIV-positive people because efficacy is not known, then there's the issue of equity amongst people that are HIV-positive.*" (South Africa-TB, TB vaccines, supply)

Other data requirements for implementation were information on coverage rates and reasons for vaccine hesitancy, cost effectiveness of the vaccines, and acceptability and feasibility of TB vaccines among the public. Finally, interviewees expressed the need for information on the epidemiology of TB disease, latent TB infection and non-tuberculous mycobacteria at the sub-national level, as it was postulated that BCG'S efficacy could differ across regions.

**India.** In terms of acceptability, interviewees expected there to be great interest in a TB vaccine at the individual level, especially in high burden areas. In terms of feasibility, interviewees anticipated that India could benefit from increased awareness of TB vaccines for adults, particularly among the elderly and building on lessons learned from COVID-19.

"*. . .because now we are doing it (vaccination campaign) for COVID-19, so the health system is not going to be surprised when we say 'OK, tomorrow, a TB vaccine is going to be there and let's do it the same way we are doing for the COVID-19 vaccine'. So it will be less damaging or less troublesome for the states. They'll (people working in the health system) be well-prepared to take it up.*" (India-TB, vaccines, policy)

In terms of finance, stakeholders indicated that the price of vaccines should be low to ensure the affordability of vaccinating the large population in India. Interviewees valued the fact that TB medical care is free in India and hoped that TB vaccines would be provided for free as well.

Interviewees were reluctant to use BCG in adolescents and adults, because previous trials conducted among Indian children found no effects. Therefore, despite familiarity with BCG, interviewees expressed more interest in an $M72/AS01_E$ -like vaccine. However, there were concerns about side effects:

"*Whenever some trial-related side effects or some effects or reactions are reported in the newspapers and news channels, sometimes they blow the issue out of proportion before knowing*

*what the real cause is. They directly relate that reaction to the vaccine. So sometimes that kind of news will create a negative impact on the trial as well and that usually happens. That's the major reason why people are a bit concerned about those vaccine trials."* (India-TB, vaccines, academic)

Interviewees anticipated that mass campaigns would not result in increased uptake, because such campaigns could be stigmatizing and result in low coverage. In terms of practical requirements to increase awareness of TB disease and uptake of a vaccine, interviewees suggested to have a communication strategy and educational campaigns:

*"The people of India are not aware. And so, we will have to educate them and, of course, we do not have a revaccination policy here. Also, we have to do advocacy, we have to adopt into the policy, based on evidence."* (India-TB, regional, policy)

Furthermore, community engagement, advocacy and the involvement of opinion leaders could ensure uptake. Since there is no adult vaccination strategy in India, this would need to be set up. However, many other barriers existed when it came to the uptake of vaccinations, not least distance to health centres. Indian interviewees were more reluctant to do targeted mass campaigns, as they worried the uptake would be low because of this stigma towards TB.

There were concerns over new infrastructure which may be needed, such as cold chain requirements, needed for M72/AS01$_E$, which were expected to be expensive:

*"If it is very difficult to maintain, and the problem with cold storage is that it will have to remain and that's very expensive. The initial storage on receiving it from wherever it is manufactured, the cold storage during the transportation, getting periodic checks to ensure the coach is not broken."* (India-TB, industry, policy)

Interviewees emphasized that any new vaccine implementation would require good adverse events monitoring and surveillance training for staff. Adverse events reporting was deemed highly important, and experts emphasized that the provider should not be blamed for the adverse event should this occur.

For targeting M72/AS01$_E$ vaccines, interviewees opted that since currently available phase IIB data only covered Mtb positives, IGRA testing may be needed, however such testing was considered infeasible and very costly:

*"Latent TB for seven million people, doing an IGRA and then providing the regime, is beyond the financial (barrier) [. . .] the numbers are so huge that they may not go with IGRA, and a standardized tuberculin is not available in India. So, we don't have an option of tuberculin as well."* (India-TB, delivery, academic)

In India, reaching male workers is a challenge as they are usually missed in vaccination campaigns conducted during their work hours. Interviewees suggested that vaccines should also be freely available at health facilities to facilitate catch up vaccination for those missing mass campaigns. It was suggested that the immunization network should expand to TB, as had happened with the SARS-CoV-2 vaccine, where different vaccines could be allocated into separate sessions on different days in the same region. Such implementation could be quick as the infrastructure is already in place.

**China.** In China, interviewees expected there to be a high acceptability of a new TB vaccine if an efficacious vaccine became available. At an individual level, the acceptability was

anticipated to be higher among people who have/had TB, and better than historically in the general population because of improvements in vaccine acceptance due to the COVID-19 pandemic:

"... *after COVID-19 pandemic, there's a very high interest among the population regarding infectious diseases. And they usually receive calls from people enquiring ways to get vaccinated.*" (China-TB, academic, policy)

Interviewees valued the familiarity with BCG, and anticipated uptake would be higher because of this. However, interviewees also looked forward to a new vaccine, with higher effectiveness and of use for adolescents and adults. Vaccine efficacy of at least 50% was considered a minimum requirement. Interviewees emphasized that the success of a novel intervention is dependent on the willingness of the government. Acceptability at the individual level would rely on the cost and level of protection. If vaccines are free and can offer high protection, there would likely be more acceptance.

Some interviewees expressed worries over low vaccine willingness as TB is a relatively rare disease in some regions and there might be a lack of awareness of the disease. However, regions with higher socio-economic status were expected to have higher vaccine willingness:

"*In eastern China where the socioeconomic status tends to be higher, the willingness and acceptability (of a new TB vaccine) might be higher in that area.*" (China-TB, vaccines, academic)

Interviewees worried that the public might doubt the effectiveness of the BCG vaccine, because although the BCG coverage rate is high in China, TB is still very prevalent. There might be less willingness and negative sentiment towards another BCG vaccination for adults. In addition, regarding mass campaigns, interviewees were concerned about potential stigma.

"*The reason why only TB-positive people will seek vaccinations is that they want to stop the development of the disease. For those who are tested negative, they would not call up and look for a vaccine or anything.*" (China-TB, academic, policy)

Interviewees worried about the lack of urgency and political will for adult and adolescent vaccines because children are generally prioritized for new vaccines. Interviewees expected greater acceptability of vaccination if the vaccine is included in the national immunization program. Stigma or hesitancy about vaccines (and side effects) need to be addressed, and proper health education and community engagement needs to be done prior and during implementation. There will be a need for health promotion because of the lack of interest in TB. Interviewees proposed acceptability and feasibility studies among target groups.

In terms of vaccination strategies, interviewees expected mass vaccination to be difficult, because of the high workload and the costs associated. They also raised that it would be challenging to vaccinate the adult population as a mass campaign. Finally, mass vaccination must be approved by the Chinese Government and approval is usually only granted during an outbreak.

"*It (mass campaign) depends on your target population and the purpose of why you are doing it. To gradually introduce this vaccine into the population, then we should definitely choose a routine vaccination. But if we are looking for short-term full coverage, then mass vaccination should be a solution.*" (China-TB, academic, policy)

In terms of logistics, interviewees highlighted that there is currently a good procurement and supply chain management structure in place in China. However, this would need to expand for adult vaccines. Logistical requirements for implementation of new vaccines named were the increase in staff workload, increased capacity of vaccine storage space, and the need for a strict protocol. Storage conditions were considered important, and storage at -80°C would be a deal breaker. Interviewees also mentioned the number of doses could be problematic, as the need for multiple doses could decrease coverage. Other potential logistic barriers may include storage space and greater staff workload.

> *"Logistic issues should not be a problem because China has a very comprehensive cold chain system and vaccination clinics–vaccination clinics are covered nationwide. I mean unless this vaccine needs very specific requirement, for instance, they need minus 80 degrees of storage, then that can be an issue, but general logistically should be fine."* (China-TB, academic, policy)

Currently, there are two types of vaccine registration in China, Category 1 vaccine (included in the national immunization program, free-of-charge vaccines) and Category 2 vaccines (not included in the national immunization program, out-of-pocket vaccines). If the new vaccine is a Category 2 vaccine, it could introduce a financial barrier for individuals. This could also cause regional differences between high-and low-income areas. Regardless of whether the vaccine will be out of pocket costs or paid by the health system, interviewees emphasized the vaccine should be low cost.

Interviewees expected large manufacturing capacity for China, and emphasized the need for product guarantee, i.e., that there is sufficient supply of $M72/AS01_E$. Overall, China has a comprehensive vaccination system. Interviews valued the experience with vaccinating adults and adolescents in influenza, HPV and SARS-CoV-2 vaccination programs, but surveillance and monitoring systems need to be scaled up to ensure adequate monitoring of vaccine safety.

## Section 2: Possible implementation scenarios

We identified on average 30 potential implementation strategies for a $M72/AS01_E$-like candidate, and 13 for a BCG revaccination-like candidate, mentioned by interviewees (Table 1). Overall, adolescents were mentioned most often as a likely target group for both vaccines (weighted average number of mentions = 5.3 and 5.7 for $M72/AS01_E$ and BCG, respectively). For $M72/AS01_E$ this was mostly driven by the results from South Africa, whereas the most frequently mentioned for $M72/AS01_E$ in China was older adults, and for India was general adults and biological high-risk groups.

There were differences in the frequency vaccination strategies proposed between the two vaccines. For $M72/AS01_E$, overall, socially vulnerable groups were the second most frequently mentioned (5.0 mentions), followed by general adults, older adults, and PLHIV. However, older adults were not mentioned by South African interviewees. For BCG revaccination, adults in the general population and older adults were the joint-second most mentioned (1.5 mentions), followed by healthcare workers and high-risk contacts of cases. However, in South Africa, apart from adolescents, health care workers were the only other group mentioned.

Table 2 shows more details of the potential implementation strategies suggested by the interviewees for an $M72/AS01_E$-like candidate. Vaccinating adolescents could be delivered through schools, and/or integrated into the current routine immunization system for adolescents and could piggyback on current HPV vaccination in China and South Africa. For $M72/AS01_E$, socially vulnerable populations were often named as a combination of people with low

**Table 1. Number of times scenarios/strategies targeting specific groups with M72/AS01$_E$ and BCG revaccination-like candidates, were mentioned by interviewees, for China, India, and South Africa.**

| M72/AS01$_E$ | China | India | South Africa | Weighted average |
|---|---|---|---|---|
| Adolescents[†] | 4 | 2 | 9 | 5.3 |
| Socially vulnerable groups* | 6 | 1 | 7 | 5.0 |
| General adults | 5 | 5 | 2 | 3.9 |
| Older people | 7 | 4 | 0 | 3.6 |
| PLHIV | 5 | 3 | 3 | 3.7 |
| Biological high-risk group** | 3 | 5 | 1 | 2.8 |
| High risk contacts*** | 3 | 3 | 0 | 1.9 |
| Health Care workers | 1 | 2 | 2 | 1.6 |
| Children | 3 | 1 | 0 | 1.4 |
| Other # | 2 | 1 | 0 | 1.0 |
| Total | 39 | 27 | 24 | 30.3 |
| BCG revaccination | China | India | South Africa | Total |
| Adolescents[†] | 5 | 5 | 7 | 5.7 |
| General adults | 2 | 3 | 0 | 1.5 |
| Older people | 2 | 3 | 0 | 1.5 |
| Health Care workers | 2 | 1 | 1 | 1.4 |
| High risk contacts*** | 3 | 1 | 0 | 1.4 |
| Socially vulnerable groups* | 2 | 0 | 0 | 0.7 |
| Biological high-risk group** | 0 | 2 | 0 | 0.5 |
| Total | 16 | 15 | 8 | 12.8 |

†Suggested age ranged from 10.-13-year-olds, 10–15, 12–15 years old, 14-year-olds, 15-year-olds, 10–19-year-olds, > 14 y/o, 12–18, 14, 14–18, 12–14, 16–18

* People with low socioeconomic status, prisoners, drug users, people living in high density areas, slums, migrants, vulnerable groups, alcoholics, miners, poor people

** diabetics, immunocompromised, comorbid patients, people in chronic care

*** TB and HCW contacts, cured TB patients

# High resource areas and occupational groups

socio-economic status, living in slums, migrants, alcoholics, unemployed men, drug users, prisoners, people living in high-risk areas and high-density populations. In India stakeholders indicated vaccination of socially vulnerable populations could be combined with free check-ups provided by the government. All countries mentioned health visits as point of care for vaccination, and South Africa mentioned workplace vaccination. Another proposed strategy was targeting the general adult population. Most interviewees suggested this to be routinely done, but in South Africa some interviewees suggested mass campaigns for M72/AS01$_E$ as well, although they could prove to be very problematic because of limited previous experience with mass campaigns in adults and older adults. Implementation of M72/AS01$_E$ for PLHIV or with co-morbid conditions could be done through the ART or other relevant condition management programs and could be implemented quickly. In China the anticipated coverage for PLHIV, through local hospitals where they receive their treatment, was considered high. In India coverage among PLHIV was anticipated to be high due to the good collaboration between the TB and HIV programs.

Table 3 shows potential implementation strategies for BCG revaccination. Adolescents were also named most likely target group and implementation could happen similarly as described for M72/AS01$_E$. Chinese and Indian interviewees suggested making this BCG vaccination compulsory at the entrance of high school, which could increase coverage. In South Africa lower anticipated coverage was ascribed to the limited availability of nurses available to

Table 2. Strategies to reach adult and adolescent populations with an M72/AS01$_E$-like candidate.

| Target population | | China | India | South Africa |
|---|---|---|---|---|
| Adolescents[†] | Implementation | Mitigate this routinely into a school-based immunization program- piggyback to HPV. | School based mass vaccination program and community based. | Routine or mass vaccination program. Mitigate this into school-based immunization program- piggybacking to HPV. |
| | Expected to start | Likely to be started ranging from 6 months to 5 years. | Could be implemented within 6 months | Could be implemented straight away to 2 years. |
| | Coverage | Ranging 30–100%, achieved within months to 8 years. | Ranging 80–85%, achieved within 1–2 years. | Ranging 30% to 95%, achieved within 1–2 years |
| Socially vulnerable groups | Implementation | Low SES, prisons, drug users and high-density population: Routine and mass vaccination upon health visit and through outreach. | Slums, migrants, vulnerable groups, alcoholics: Mass vaccination through periodic screening. | Migrants, men (unemployed), miners, poor people, people living in high-risk area: Routine and mass vaccination upon health visit and trough workplace |
| | Expected to start | Could be implemented right away | Could be implemented 6 months- 1 years | Could be implemented within months to 2 years |
| | Coverage | Coverage ranging from 5 to 40%, achieved short term. Strategy needs to be overlapped with 6 | Coverage ranging 50–80%, achieved in 2 years. | Coverage ranging 50–95%, achieved in 2 years |
| General adults | Implementation | Routine vaccination at community level, outreach, or primary care level. | Routine vaccination through adult's vaccination program, outreach, or primary care level. | Mass vaccination at clinics, |
| | Expected to start | Could be implemented right away for 3 years. | Could be implemented right away to 6 months—1 year. | Implementation within 1 year. |
| | Coverage | Coverage ranging 20–50%, achieved within 3–10 years. | Coverage ranging 10–90%. | Coverage 40–60%, within 1–3 years |
| Older people** | Implementation | Routine vaccination through TB screening at community clinic years | Routine upon health visit or mass vaccination through outreach and primary care or private | Not mentioned |
| | Expected to start | Could be implemented right away to 6 years | Could be implemented 6 months- 1 years | |
| | Coverage | Coverage ranging 30–95%%, achieved within 1–10 years | Coverage ranging 10–65%, achieved within 1–10 years | |
| PLHIV | Implementation | Routine at visit art local treating hospital for ART or mass vaccination at outreach through screening, both through ART program | Routine at visit local treating hospital for ART or mass vaccination trough ART program | Routine at visit art local treating hospital for ART |
| | Expected to start | Could be implemented right away to 5 years | Could be implemented 3 months to 1 year | Could be implemented 6 months to 2 years |
| | Coverage | Coverage ranging 5–100%, achieved within 1–5 years. | Coverage ranging 70–95%, achieved within 6 months to 1 year. | Coverage ranging 60–80%, achieved within 6 months to 1 year. |
| Biological high-risk group | Implementation | 4.c Diabetics, immunocompromised: Routine vaccination upon health visit. | Comorbid patients, Diabetics, malnourished, low BMI. Mass vaccination through outreach based on priority list per districts and have phases months. | People in chronic care (egg diabetes and hypertension): Routine vaccination upon health visit. |
| | Expected to start | Could be implemented 3–4 years. | Could be implemented 6 months- 2 years. | Could be implemented 18 months- 2 years. |
| | Coverage | | Coverage ranging 40–80%, achieved in 2 years. | Coverage between 60–70%, achieved in 2 years |
| Health Care workers | Implementation | Routine vaccination at vaccine clinic. | Mass vaccination at clinic. | Routine vaccination through vaccination program. |
| | Expected to start | Could be implemented 6 months after policy recommendation/ registration. | Could be implemented within 3 months. | Could be implemented 1–2 years. |
| | Coverage | Coverage 70–80%, within 2–3 years. | Coverage 50%, within 1 years. | Coverage ranging from 60–90%, within 3 years. |

(*Continued*)

**Table 2.** (Continued)

| Target population | | China | India | South Africa |
|---|---|---|---|---|
| **High risk contacts** | *Implementation* | TB and HCW contacts: Mass vaccination arranged through contact investigation. | TB household contacts and cured TB patients: Routine and mass vaccination through clinical follow up and contact investigation at primary care. | Not mentioned |
| | *Expected to start* | | Could be implemented within 3–5 months | |
| | *Coverage* | | Coverage ranging from 60–70%, achieved 10 months-1 years | |

PLHIV = people living with HIV; SES = socioeconomic status; HCW = health care workers; ART = antiretroviral treatment; HPV = human papilloma virus

† = Suggested age ranged from 6–18, 10–13, 8–12

** = seniors, 65+, old age groups

go to high schools. Present strategy limits to school-going adolescents. BCG was not considered as a feasible option for PLHIV in all countries, nor in the general adult population in South Africa.

For both vaccines, implementation could take some time to prepare. With substantive and concentrated effort, the time until implementation could be quick, and coverage could be high. Interviewees emphasized that determining the target population and frequency of mass vaccination campaigns would depend on the efficacy of the new vaccine. An additional strategy included screening the whole population with an IGRA test and then giving the M72/AS01$_E$ like vaccine to IGRA positive and BCG revaccination to IGRA negatives.

## Discussion

We assessed the acceptability and feasibility of implementing TB vaccines for adolescents and adults among key interviewees from China, India, and South Africa. Tuberculosis vaccines targeting adolescents and adults were of interest in all three countries, with school-aged adolescents identified most likely a target population in most settings. Proposed implementation strategies varied by country, likely due to demographic and epidemiological differences, for example older people were highlighted as target group in China and India. Due to stigma and logistical challenges with focused mass campaigns, routine vaccination was favoured in all countries. Interviewees mentioned that there is a bottleneck in TB control and novel vaccines are needed.

Overall, there was anticipated high political will in the three countries for novel TB vaccines, because of the high TB burden. The expected efficacy in individuals with *Mycobacterium tuberculosis* (Mtb) infection and efficacy for people living with HIV was a perceived benefit for M72/AS01$_E$ implementation. A new infrastructure and the two-dose regimen necessary for M72/AS01$_E$ were seen as obstacles. Interviewees valued that we are already knowledgeable with BCG but disliked the contraindication for PLHIV (especially in South Africa). Interviewees highlighted that pre-vaccination testing would be needed to target Mtb positives or negatives only, which was considered infeasible and very costly. Conducting clinical trials in a local setting, acquiring TB epidemiological data, and addressing vaccine hesitancy were suggested requirements to support implementation.

In terms of implementation strategies, adolescents were chosen as a likely target group as they could be easily reached through school-based programs, and where in place by piggybacking on current vaccine programs to ensure high coverage. In South Africa and India, school based vaccination is commonly used for HPV vaccination and affords good access [15].

**Table 3. Strategies to reach adult and adolescent populations with a BCG revaccination-like candidate.**

| Target population | | China | India | South Africa |
|---|---|---|---|---|
| **Adolescents*** | *Implementation* | Mitigate this routinely into school clinics, compulsory upon enrolment. | Schools based mass campaign or annual routine vaccination at health system check-up or HPV., | Routine or mass vaccination program. Mitigate this into school-based immunization program- piggyback to HPV. |
| | *Expected to start* | Likely to be started ranging from 6 months to 3 years. | Could be implemented within 3–6 months | Could be implemented straight away to 2 years. |
| | *Coverage* | Ranging from 30% coverage in 5–10 years for non-compulsory strategy to 50–100% for compulsory within 1 year | Ranging 70–85%, achieved within 2 years. | Ranging 30% to 100%, achieved within 6 months to 2 years |
| **General adults** | *Implementation* | Routine vaccination or mass in combination with screening | Routine vaccination through adult vaccination program or mass through outreach | Not mentioned |
| | *Expected to start* | Could be implemented right in 6 months | Could be implemented right away for 6 months—1 year. | |
| | *Coverage* | Coverage ranging 30–90%, achieved within 3–8 years. | Coverage ranging 60–95%, achieved within 2 years | |
| **Older people**** | *Implementation* | Routine vaccination (IGRA-) through TB screening at community clinic years and yearly national senior exams. | Routine through adult vaccination program or mass vaccination through outreach or primary care | Not mentioned |
| | *Expected to start* | Could be implemented 6 months to 1 year | Could be implemented within 6 months to 1 year | |
| | *Coverage* | Coverage ranging 30–95%%, achieved within 1–10 years | Coverage ranging 30–95%%, achieved within 2 years | |
| **Health Care workers** | *Implementation* | Routine vaccination at vaccine clinic. | Not mentioned | Routine vaccination through vaccination program, piggyback to influenza/ Hepatitis B |
| | *Expected to start* | Could be implemented 6 months | | |
| | *Coverage* | Coverage 70–80%, within 2–5 years. | | Coverage 60–70%, achieved shortly |
| **High risk contacts** | *Implementation* | TB and HCW contacts: Mass vaccination upon needs through outreach or routine vaccination | TB household contacts: routine vaccination upon health visit to local primary care. | Not mentioned |
| | *Expected to start* | | Could be implemented within 3 months | |
| | *Coverage* | Coverage 70–80% | Coverage 60–70%, within 1 year | |
| **Socially vulnerable groups** | *Implementation* | Prisons, and high-density population: Routine vaccination | Vulnerable population: socially vulnerable; slums; migrants; Mass vaccination trough periodic screening | Not mentioned |
| | *Expected to start* | | Could be implemented within 6 months | |
| | *Coverage* | | Coverage 80%, within 2 years | |
| **Biological high-risk group** | *Implementation* | Not mentioned | Vulnerable population: people with a comorbidity / non-communicable disease/ low BMI/Diabetics/ PLHIV / alcoholics: Mass vaccination, through active case finding | Not mentioned |
| | *Expected to start* | | Could be implemented within 6 months to 1 year | |
| | *Coverage* | | Coverage 50–80%, within 2 years | |

PLHIV = people living with HIV; SES = socioeconomic status; HCW = health care workers; ART = antiretroviral treatment; HPV = human papilloma virus.

* Suggested age ranged from: college students, 10.-13-year-olds, 10–15, 12–15 years old, 14-year-olds, 15-year-olds, 10–19-year-olds, > 14 y/o, 12–18, 14, 14–18, 12–14, 16–18

** Older people = Seniors 60+ and 'elderly' 65+; older age groups, older people and older people with diabetes

The high HIV incidence rate in South Africa, estimated at 357 per 100 000 in 2019 [16], made stakeholders likely more reluctant to use BCG. While HIV is less of a problem in India and China, reluctance towards BCG in India could be explained by previous trials in infants that did not show beneficial effects [17]. For BCG, South African interviewees emphasized the need for an HIV test prior to vaccination, which could decrease the feasibility of implementation. Socially vulnerable groups were a likely target group in India and South Africa for M72/AS01$_E$, due to the high burden of TB among these groups. Bazargan et al. [18] also highlighted that minorities are a commonly seen group to target for influenza vaccines. In China, older people were frequently named as a target group, while in South Africa this group was not mentioned. This was likely due to demographic and epidemiological differences, whereas the rate in elderly relative to young adults is higher in China than SA. In China 12% of the population is above 65 years of age [19], as compared to 5.5% in South Africa [20].

Interviewees considered efforts need to be undertaken to address practical requirements such as gaps in monitoring systems in current infrastructure. Grignard et al. [21] also highlight logistical and infrastructure requirements needed to be addressed prior to implementation and addressing the issue of scale up in terms of human and technical resources is essential. Similarly, they also highlight the issue of education on TB disease and vaccination and the need for communication campaigns and community engagement, which was also mentioned by our interviewees. Interviewees suggested lessons learned from COVID-19 vaccines strategies would help to inform TB vaccination strategies. In terms of evidence, interviewees from all countries required safety data, efficacy, and cost effectiveness of the vaccines.

Our findings align with Gallagher et al. [22] and McClure et al. [23] on the importance of addressing hesitancy and rumours, as this can have a negative impact on vaccination programs. China and India indicated studies in the local context were needed, as was evidence demonstrating the safety and efficacy in target groups. Interviewees expressed the need for more long-term country (or even regional) specific epidemiological data such as infection and disease prevalence and birth rate, to evaluate which vaccine should be targeted were. These findings align with the systematic review by Gates et al. [24] in terms of the importance of safety requirements vaccines in general and anticipated uptake for adult vaccines. In addition, the infrastructure in place for delivering vaccines is substantially different between countries, emphasizing the importance of obtaining country-specific requirements. Cost and affordability were deemed important to ensure equity of vaccine distribution. Giersing et al. [25] present similar requirements for vaccine delivery of novel vaccines such as: efficacy, safety, cost effectiveness, cold chain and logistical issues, acceptability, and that lower vaccine costs could increase vaccine uptake. Although the Giersing et al. assessment was limited to the expanded immunization program for children. For mass vaccination, interviewees expected that a lower coverage would be achieved for the second dose of the vaccine. This concept of reduced coverage for a follow up dose was also seen with HPV vaccinations [15].

Our study has limitations. This research did not include any patient populations, which are important to determine acceptability and hesitancy; our focus was on policy makers and program implementers. Some of the questions in the study were hypothetical in nature, further research will be required once phase III studies have been conducted and a full understanding of vaccine characteristics is available. However, such research is important at this moment in development to ensure key populations (e.g., older adults, PLHIV, risk groups) are included in late stage trials, and allow preparation to begin for implementing such vaccines to the prioritized groups as soon as possible after phase III results are available. For the implementation scenarios section, BCG was discussed second, leaving a shorter time for this discussion. As a result of the COVID-19 pandemic, we conducted interviews using Zoom, and although most

interviews went smoothly, some technical difficulties were experienced, which could increase the chance of loss of information and misinterpretation [26]. Interviews were conducted during the pandemic, during which understanding and insights regarding adult vaccination have changed, so later interviews may have benefitted from more COVID-related insights. Input was given without the interviewees having insight about the epidemiological impact or cost-effectiveness of targeting each group, such data would likely be part of final implementation decisions and should be part of future research once such data are available. Finally, we had underrepresentation of interviewees on local level, civil society, finance, and in vaccine delivery.

This is the first study that we are aware of that evaluates adult and adolescent TB vaccine implementation, offering unique perspectives from a country-specific health system standpoint. The richness of the conversations and the results demonstrated the importance of early stakeholder participation and should be leveraged to design late-stage trials (ensuring key populations such as older adults, PLHIV and risk groups are included) and to inform implementation plans for new vaccines, including ensuring early preparation of programs for these harder to reach populations. Interviewees appreciated early consultation in defining and scrutinizing implementation strategies and possible challenges early on. With the increasing focus on vaccination to protect health at every stage in life, we may see more adult and adolescent vaccines coming through the pipeline in years to come. Future adult and adolescent vaccine implementation studies may benefit from studies using similar methods. We can learn from how countries implement their SARS-CoV-2 vaccination campaigns and gain insights into how these preferences can be put into practice when thinking about rolling out TB vaccines in similar populations with similar requirements. Our findings will inform epidemiological, cost-effectiveness and budget impact modelling, which will be the next steps in assessing feasibility for introducing these late-stage TB candidates.

## Conclusion

In all three countries, implementation challenges and benefits were recognized. Our research provides critical information on the acceptability and feasibility of novel tuberculosis vaccines in certain target groups as well as country-specific acceptability and feasibility. Key groups for vaccine deployment in these contexts have been identified, and clinical studies and implementation planning should begin immediately. This research can help inform translation from phase IIB clinical trials through to phase III and hopefully implementation of the most advanced new TB vaccine candidate, including vaccine development strategy, country-level vaccination policy, and global vaccine strategy/policy.

## Supporting information

**S1 Table. Overview of expertise of interviewees.**
(DOCX)

**S1 File. Background information.**
(DOCX)

**S2 File. Interview guide.**
(DOCX)

**S3 File. Informed consent.**
(DOCX)

## Acknowledgments

We would like to thank all interviewees in South Africa, China and India for their participation sharing their invaluable experiences and insights. Finally, we want to thank Ineke Spruijt for her review and support.

## Author Contributions

**Conceptualization:** Janet Seeley, Hisham Moosan, Richard G. White, Rebecca C. Harris.

**Data curation:** Puck T. Pelzer.

**Formal analysis:** Puck T. Pelzer.

**Funding acquisition:** Janet Seeley, Richard G. White, Rebecca C. Harris.

**Investigation:** Puck T. Pelzer, Li Tao, Zhao Yanlin.

**Methodology:** Puck T. Pelzer, Janet Seeley, Chathika Weerasuriya, Miqdad Asaria, Sahan Jayawardana, Richard G. White, Rebecca C. Harris.

**Project administration:** Puck T. Pelzer.

**Resources:** Puck T. Pelzer, Michele Tameris, Li Tao, Zhao Yanlin, Hisham Moosan.

**Supervision:** Janet Seeley, Richard G. White, Rebecca C. Harris.

**Validation:** Janet Seeley, Fiona Yueqian Sun, Michele Tameris, Hisham Moosan.

**Visualization:** Puck T. Pelzer.

**Writing – original draft:** Puck T. Pelzer, Miqdad Asaria.

**Writing – review & editing:** Janet Seeley, Fiona Yueqian Sun, Michele Tameris, Li Tao, Zhao Yanlin, Hisham Moosan, Chathika Weerasuriya, Sahan Jayawardana, Richard G. White, Rebecca C. Harris.

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
