## [Decision Letter · Decision Letter 0]

15 Nov 2021

Potential implementation strategies, acceptability, and feasibility of new and repurposed TB vaccines

PGPH-D-21-00717

Dear Dr. Ms Pelzer,

We're pleased to inform you that your manuscript has been judged scientifically suitable for publication and will be formally accepted for publication once it meets all outstanding technical requirements.

Within one week, you'll receive an e-mail detailing the required amendments. When these have been addressed, you'll receive a formal acceptance letter and your manuscript will be scheduled for publication.

An invoice for payment will follow shortly after the formal acceptance. To ensure an efficient process, please log into Editorial Manager at https://www.editorialmanager.com/pgph/ click the 'Update My Information' link at the top of the page, and double check that your user information is up-to-date. If you have any billing related questions, please contact our Author Billing department directly at authorbilling@plos.org.

Kind regards,

Nnodimele Onuigbo Atulomah, PhD

Academic Editor

Additional Editor Comments (optional):

Three reviews were positive regarding the submission. The rationale for undertaking this study is well justified considering that vaccine hesitancy is a growing challenge in every global community which has become exacerbated by misinformation resulting from the rapid development of SARS-Cov-2 vaccines driven by technological advancement in Gene technology. The authors expressed the belief that the study has relevance in preparing population of adolescents and adults towards accepting the vaccines when it is rolled out considering the encouraging results reported from ongoing clinical trials for two Phase 2B tuberculosis vaccines. The programme developed would be helpful in facilitating vaccine acceptance when implemented prior to rolling out the vaccines at the end of development. This preliminary study can provide understanding of the responses of the population and policy support for implementing those strategies that are most likely to bring about anticipated public response.

Reviewers' comments:

Reviewer's Responses to Questions

**Comments to the Author**

1. Does this manuscript meet PLOS Global Public Health’s publication criteria? Is the manuscript technically sound, and do the data support the conclusions? The manuscript must describe methodologically and ethically rigorous research with conclusions that are appropriately drawn based on the data presented.

Reviewer #1: Yes

Reviewer #2: Yes

Reviewer #3: Yes

2. Has the statistical analysis been performed appropriately and rigorously?

Reviewer #1: Yes

Reviewer #2: Yes

Reviewer #3: N/A

3. Have the authors made all data underlying the findings in their manuscript fully available (please refer to the Data Availability Statement at the start of the manuscript PDF file)?

Reviewer #1: Yes

Reviewer #2: Yes

Reviewer #3: No

4. Is the manuscript presented in an intelligible fashion and written in standard English?

Reviewer #1: Yes

Reviewer #2: Yes

Reviewer #3: Yes

5. Review Comments to the Author

Reviewer #1: The manuscript presents an assessment of potential implementation strategies, acceptability, and feasibility of new and repurposed TB vaccines among key stakeholders in South Africa, India and China using qualitative method. The manuscript is generally well written and clearly presented.

Reviewer #2: The title is good and reflects in the body of the manuscript. The abstract was also carefully written, though can be improved upon. The introduction(Background to the study) was properly drafted to give readers some information such as global statistics on the problem phenomenon and narrowed it down to the regional level.

The methodology is appropriate since its a qualitative study. However, more finding on potential implementation strategies of TB Vaccines would have been uncovered if the Authors included the target population as part of the interviewees and not only the decision makers as done in the study. The author also mentioned some key constructs in behavioural model such as perceived efficacy, perceived benefits but no proper indication on how a particular model was operationalised in the tool used for data collection. The selection of India, South Africa and China was rightly justified in the study.

Analysis of the results from different interviews were properly done.

Ethical issues was not properly addressed. Lastly, no policy implication was mentioned in the conclusion and no recommendation that could help in implementation strategies of TB vaccines, although the authors suggested for immediate clinical trials as a follow up on the study's outcome which is a very good recommendation.

Reviewer #3: Soundness of article- work is sound and provides relevant information that can aid health promotion interventions on vaccination efforts in the countries indicated and others who may be able to apply lessons learnt.

The synchrony of background with the problem and justification for the study is commendable- the background provides us a broad perspective of the problem while also pointing to the need to focus on the human target population and countries being studied. In addition, while not explicitly stated, it is clear the research paradigm on which the study is underpinned-a social constructivist perspective to understand the context for action in Public Health care and achieve a global goal.

Given the paradigm within which the study lies and over arching goal, the methodology used is appropriate to answer the research questions. The method used in data collection and procedures are appropriate. Proding expert opinion in this research shows a well thought out validation process.

The problem is well aligned and logical and also clearly indicates the gap that needs to be filled. The discussion takes us through perspectives from other researchers and clearly addresses the limitations of the study which are very germane to the study while also pointing to further studies within the broad research - which awakens the reader to expectations in further reads.

Generally, the study is well written with minimal grammatical corrections. The ancronym PLHIV is used to indicate "People Living with HIV/AIDS". This is new to me as a reader and is a deviation from the norm PLWHA. However, the authors were able to provide their definition of PLHIV and may be left as is since clearly defined.

6. PLOS authors have the option to publish the peer review history of their article (what does this mean?). If published, this will include your full peer review and any attached files.

**Do you want your identity to be public for this peer review?** For information about this choice, including consent withdrawal, please see our Privacy Policy.

Reviewer #1: **Yes: **Titilayo Olaoye

Reviewer #2: **Yes: **Mustapha Adebayo

Reviewer #3: **Yes: **Dr. Ajike Saratu Omagbemi
